# Exchangeable Neural ODE for Set Modeling

**Yang Li**[*]
Department of Computer Science
University of North Carolina at Chapel Hill
yangli95@cs.unc.edu

**Haidong Yi**[*]
Department of Computer Science
University of North Carolina at Chapel Hill
haidyi@cs.unc.edu

**Christopher M. Bender**
Department of Computer Science
University of North Carolina at Chapel Hill
bender@cs.unc.edu

**Siyuan Shan**
Department of Computer Science
University of North Carolina at Chapel Hill
siyuanshan@cs.unc.edu

**Junier B. Oliva**
Department of Computer Science
University of North Carolina at Chapel Hill
joliva@cs.unc.edu

## Abstract

Reasoning over an instance composed of a set of vectors, like a point cloud, requires that one accounts for intra-set dependent features among elements. However, since such instances are unordered, the elements' features should remain unchanged when the input's order is permuted. This property, permutation equivariance, is a challenging constraint for most neural architectures. While recent work has proposed global pooling and attention-based solutions, these may be limited in the way that intradependencies are captured in practice. In this work we propose a more general formulation to achieve permutation equivariance through ordinary differential equations (ODE). Our proposed module, Exchangeable Neural ODE (ExNODE), can be seamlessly applied for both discriminative and generative tasks. We also extend set modeling in the temporal dimension and propose a VAE based model for temporal set modeling. Extensive experiments demonstrate the efficacy of our method over strong baselines.

## 1 Introduction

The vast majority of machine learning models operate on an independent and identically distributed (i.i.d.) vector, $x \in \mathbb{R}^d$. In some cases, however, the inputs may contain a set of instances, $\mathbf{x} = \{x_i\}_{i=1}^n$, which jointly determine the target. We note that instances within a set may interact with each other. For instance, the points inside a point cloud jointly determine the global structure. In this work, we build both discriminative and generative models on sets, which explore the intradependencies within a set to capture both global and local structures.

A set is a collection of data that does not possess any inherent ordering of its elements. In statistics, a set is described as an exchangeable sequence of random variables whose joint probability distribution does not change under any permutation $\pi$, i.e.,

$$p(x_1, \ldots, x_n) = p(x_{\pi_1}, \ldots, x_{\pi_n}). \tag{1}$$

---

[*]equal contribution

Discriminative models that operate on a set must predict a target $y$ that is invariant to all permutations. Applications for such models include population statistics estimation, point cloud classification, etc. A naive approach where training data are augmented with random permutations and treated as sequences has been empirically proven insufficient [1]. Previous works [2, 3] developed simple permutation invariant operations by processing each element independently and then aggregating them using a pooling operation (max, mean, etc). However, such an operation largely ignores the intradependencies between elements within the set. In this work, we introduce an inductive bias into the model to exploit said intradependencies across elements. Specifically, we introduce a permutation equivariant module to explicitly model the dependencies among set elements.

Set generative models with tractable, exchangeable likelihoods have recently been investigated (that is, likelihoods which are invariant to permutations) [3, 4, 5]. Simple approaches that estimate likelihood for each instance independently are insufficient since global structures cannot be inferred. To overcome this shortcoming, we construct a flow-based generative model for tractable likelihood estimation on sets.

The key for both discriminative and generative set models is a powerful equivariant transformation that captures set intradependencies. In order to compute the likelihood for a flow based generative model, the transformation additionally requires to be invertible. In this work, we propose an exchangeable, invertible flow transformation, ExNODE, based on Neural Ordinary Differential Equation (NODE) [6]. Invertibility is guaranteed via the NODE framework since integration backward in time is always possible. We implement ExNODE by parametrizing a differential equation with a permutation equivariant architecture.

In addition to modeling the sets in spatial dimensions, we extend ExNODE to the temporal dimension and propose a temporal set modeling task. Such a set model has many potential applications, including modeling the evolution of galaxies, pedestrian tracking, etc. Here, we utilize a VAE-based framework with our proposed set generative model as the decoder. The temporal evolution is captured by another ODE in the latent space. After training, our model can interpolate and extrapolate to generate sets at unseen (potentially fractional) time steps.

Our contributions are as follows: 1) We propose ExNODE, an exchangeable module for set modeling, which explicitly captures the intradependencies among set elements. 2) ExNODE represents a type of invertible flow transformation on which the invariant set likelihood can be achieved. 3) We propose a temporal set modeling task and a VAE-based model for time variant set modeling. The temporal VAE utilizes differential equations to transit hidden states in time. To the best of our knowledge, our model is the first one designed for temporal sets. 4) We achieve state-of-the-art performance for both point cloud classification and likelihood estimation.

## 2 Background and Related Works

### 2.1 Set Modeling

A set is a collection that does not impose ordering among its elements. Models over sets *must* preserve this property. We list the commonly used terminology for set modeling below. We denote a set as $\mathbf{x} = \{x_i\}_{i=1}^n \in \mathcal{X}^n$, where $n$ is the cardinality of the set and the calligraphic letter $\mathcal{X}$ represents the domain of each element $x_i$.

**Definition 1.** *(Permutation Equivariant) Let $f : \mathcal{X}^n \to \mathcal{Y}^n$ be a function, then $f$ is permutation equivariant iff for any permutation $\pi(\cdot)$, $f(\pi(\mathbf{x})) = \pi(f(\mathbf{x}))$.*

**Definition 2.** *(Permutation Invariant) Let $f : \mathcal{X}^n \to \mathcal{Y}$ be a function, then $f$ is permutation invariant iff for any permutation $\pi(\cdot)$, $f(\pi(\mathbf{x})) = f(\mathbf{x})$.*

A naive method to encourage permutation invariance is to augment the training data with randomly permuted sets and treat them as sequences. One could then train a neural network mapping the permuted inputs to the same output. Due to the universal approximation ability of neural network, the final model could be invariant to permutations given an infinite amount of training data and model capacity. However, this simple approach does not guarantee invariance for real-world finite datasets. As pointed out in [1], the order cannot be discarded for a sequence model.

DeepSet [2] proves that any permutation invariant function for a set with finite number of elements can be decomposed as $\rho(\sum_{x \in \mathcal{X}^n} \phi(x))$, where the summation is over the set elements. Based on this

decomposition, they propose using two neural networks for both $\rho$ and $\phi$ to learn flexible permutation invariant functions. They also propose an equivariant model, where independent processing combined with a pooling operation is used to capture the intradependencies. Although deep sets are universal approximators, there are some constraints with respect to the dimensionality of latent representation as shown by [7].

Set Transformer [8] proposes to use an attention mechanism over set elements to model the intradependencies between each pair of elements. Since the attention is a weighted sum over all set elements, this operation is naturally equivariant. They also propose an attention based pooling operation to achieve invariant representations.

Set likelihood estimation requires the likelihood to be invariant to permutations, i.e.

**Definition 3.** *(Exchangeable Likelihood) Given any permutation $\pi$ and finite random variables $x_i, i = 1, \ldots, n$, then the likelihood of $\{x_i\}_{i=1}^n$ is exchangeable iff*

$$p(x_1, \ldots, x_n) = p(x_{\pi_1}, \ldots, x_{\pi_n})$$

A naive approach might be to estimate the likelihood independently for each element. Neural Statistician [3] utilizes a VAE-based model inspired by the de Finetti's theorem, where conditionally independent likelihoods are estimated for each element by conditioning on a permutation invariant latent code. PointFlow [9] extends the Neural Statistician by using normalizing flow for both the encoder and decoder. Both flow models operate independently on each element. BRUNO [4] employs an independent flow transformation for each element and an exchangeable student-t process for the invariant likelihood. FlowScan [5] transforms the set likelihood problem to the familiar sequence likelihood problem via a scan sorting operation. In this work, we extend a flow based generative model for exchangeable sets with a tractable invariant likelihood.

Modeling sets is an example of a bigger family of models that integrate prior knowledge about the underlying data distribution. Some existing works [10, 11, 12] have also explored the equivariant and invariant properties, but they are not built particularly for sets.

## 2.2 Neural ODE

Connection between neural networks and differential equations has been studied in [13, 14], where classic neural network architectures are interpreted as discretizations of differential equations. Built upon those works, [6] proposed the *Neural ODE*, which employs the adjoint method to optimize the model in a memory efficient way. Based on the connection between ResNet [15] and Euler discretized ODE solver, they propose to use other ODE solvers to implicitly build more advanced architectures. A basic formulation of Neural ODE is shown as:

$$\frac{dh(t)}{dt} = f_\theta(h(t), t), \quad h(t_0) = x, \tag{2}$$

where $f_\theta$ is parametrized as a neural network. Neural ODE blocks process input $h(t_0)$ using a black-box ODE solver so that

$$h(t_1) = h(t_0) + \int_{t_0}^{t_1} f_\theta(h(t), t) dt. \tag{3}$$

Neural ODE represents a type of continuous depth neural network. Comparing with discrete depth neural networks, Neural ODE has several advantages: 1) In theory, there could be infinite number of layers that share the same set of parameters. Hence, Neural ODE is more parameter efficient. 2) The gradients w.r.t. $\theta$ can be computed using adjoint method [6] that only requires $O(1)$ memory usage, since the intermediate variables do not need to be stored during forward pass; they can be recovered during back propagation. 3) Neural ODE is naturally invertible if $f_\theta$ satisfies certain conditions, such as Lipschitz continuity.

## 2.3 Continuous Normalizing Flow (CNF)

Normalizing flows (NFs) [16, 17, 18] are a family of methods for modeling complex distributions in which both sampling and density evaluation can be efficient and exact. NFs use the change of variable theorem to calculate the likelihood of training data:

$$\log p_{\mathcal{X}}(x) = \log p_{\mathcal{Z}}(z) + \log \left| \det \frac{\partial q(x)}{\partial x} \right|, \tag{4}$$

where $p_{\mathcal{X}}(x)$ is the likelihood in input space, $p_{\mathcal{Z}}(z)$ is the likelihood evaluated on a base distribution, and $z = q(x)$ is an invertible transformation which transforms inputs to latent space. The base distribution is typically chosen as a simple distribution such as isotropic Gaussian. To allow efficient likelihood evaluation, NFs typically employ transformations $q(\cdot)$ with a triangular Jacobian so that the determinants can be computed cheaply, although it reduces the flexibility and capacity of NFs.

[6] and [19] propose continuous normalizing flows (CNFs) and extend the change of variable theorem to continuous-time case:

$$\frac{d \log p(z(t))}{dt} = -\operatorname{Tr}\left(\frac{\partial f}{\partial z(t)}\right), \tag{5}$$

where $\frac{dz}{dt} = f(z(t), t)$ is a differential equation describing the dynamics of $z(t)$ as in Eq. (2). Unlike in Eq. (4) where variables are transformed explicitly by $q$, CNF implicitly transforms the variables by integration, i.e.,

$$q(x) = z(t_1) = z(t_0) + \int_{t_0}^{t_1} f(z(t), t)dt, \quad x = z(t_0). \tag{6}$$

Eq. (5) requires only the trace of Jacobian matrix rather than the more expensive determinants in Eq. (4), which reduces the computation complexity dramatically. As a result, CNFs can afford using more flexible transformations implicitly implemented by integrating $f$. Equation (5) also indicates that the change of log density is determined by another ODE that can be solved with $z(t)$ itself simultaneously using an ODE solver.

## 3 Method

In this section, we introduce the permutation equivariant module, ExNODE. We discuss how to apply ExNODE for different set modeling tasks. We consider both discriminative (set classification) and generative (set generation with flow models) tasks. Finally, we explore temporal set modeling.

### 3.1 Exchangeable Neural ODE

Our permutation equivariant module for exchangeable sets is based on differential equations. Specifically, we can prove the following theorem. The detailed proof is provided in Appendix A.

**Theorem 1.** *(Permutation Equivariant ODE) Given an ODE $\dot{\mathbf{z}}(t) = f(\mathbf{z}(t), t), \mathbf{z}(t) \in \mathcal{X}^n$ defined in an interval $[t_1, t_2]$. If function $f(\mathbf{z}(t), t)$ is permutation equivariant w.r.t. $\mathbf{z}(t)$, then the solution of the ODE, i.e., $\mathbf{z}^\star(t), t \in [t_1, t_2]$ is permutation equivariant w.r.t. the initial value $\mathbf{z}(t_1)$. We call the ODE with permutation equivariant properties ExODE.*

Following Neural ODE [6], we parametrize $\dot{\mathbf{z}}(t)$ with a neural network. To ensure the integrated function $\mathbf{z}^*(t)$ is permutation equivariant, we build $\dot{\mathbf{z}}(t)$ in a permutation equivariant form. Specifically, $f$ is implemented as a permutation equivariant neural network, such as the deepset equivariant layer or the attention based set transformer layer.

An additional benefit of our ExNODE is its invetibility. Since we can always integrate from $t_2$ to $t_1$, it does not require any special design as in typical flow models to guarantee invertibility. Therefore, our ExNODE can be easily plugged into flow models as a transformation. According to Eq. (5), the likelihood can be similarly evaluated.

In order to guarantee the initial value problem have unique solution, the dynamic $f(\cdot)$ needs to be Lipschitz continuous. However, [20] proves that the dot-product self attention module is not Lipschitz and propose a L2 formulation of the attention module with finite Lipschitz constant. We leave it to future work to explore this L2 self-attention. In this work, we instead bound the inputs by normalizing them to the range $[0, 1]$ to make sure the dynamics are Lipschitz continuous, since any continuously differentiable function is Lipschitz within a compact input space [20].

### 3.2 Set Classification

For the set classification task, a model must guarantee that the order of set elements does not affect the prediction results. Hence, given a set $\mathbf{x} = \{x_1, \ldots, x_n\} \in \mathcal{X}^n$, our purpose is to learn a permutation invariant function that maps $\mathbf{x}$ to its corresponding label $y$.

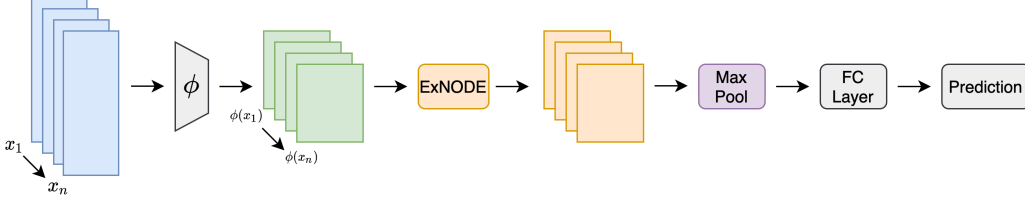

Figure 1: Illustration of the architecture of our set classification model. The function $\phi(\cdot)$ refers to independent operations that expand the dimension. The *ExNODE* may contain multiple ODE blocks. The max pooling is applied across set elements.

Notice that multiple equivariant layers stacked together are overall equivariant, we employ a permutation invariant architecture by stacking multiple equivariant layers and a pooling aggregating operation. Figure 1 illustrates the architecture of our set classification model. First, we use a linear mapping $\phi$ to expand the feature dimensions independently for each set element. Then, permutation equivariant ODEs serve as a dimension-preserving nonlinear mapping to capture the dependencies among set elements and learn the feature representations for $\mathbf{x}$. When feature representations are available, we use a max pooling to aggregate the information across $x_i$. After max pooling, we get a permutation invariant vector representation that summarizes the set $\mathbf{x}$. We denote the embedding vector as $v$,

$$v = \text{MaxPool}(\text{ExNODE Solve}(\phi(\mathbf{x}))). \tag{7}$$

Finally, we use fully connected (FC) layers and a softmax layer to predict labels $y$.

### 3.3 Continuous Normalizing Flow for Sets

We extend the continuous normalizing flow proposed in [6, 19] to model exchangeable sets $\mathbf{x} \in \mathcal{X}^n$. Specifically, we have the following proposition from [5], repeated here for convenience:

**Proposition 1.** *For a flow model with transformation $q(\cdot)$ and base likelihood $p_{\mathcal{Z}}(\cdot)$, the input likelihood $p_{\mathcal{X}}(\mathbf{x}) = p_{\mathcal{Z}}(q(\mathbf{x})) \left| \det \frac{dq}{d\mathbf{x}} \right|$ is exchangeable if the transformation is permutation equivariant and the base likelihood is invariant.*

Similar to Eq. 6, we parametrize transformation $q$ implicitly as a differential equation, i.e.,

$$\dot{\mathbf{z}}(t) = f_\theta(\mathbf{z}(t), t), \quad \mathbf{z}(t_0) = \mathbf{x}, \tag{8}$$

where $f_\theta$ is a permutation equivariant neural network w.r.t. $\mathbf{z}(t)$. Using the instantaneous change of variables formula, the log likelihood of $\mathbf{z}(t_1)$ and $\mathbf{z}(t_0)$ satisfy the following equation:

$$\log p(\mathbf{z}(t_0)) = \log p(\mathbf{z}(t_1)) + \int_{t_1}^{t_0} \text{Tr}\left( \frac{\partial f_\theta}{\partial \mathbf{z}(t)} \right) dt, \tag{9}$$

where $\mathbf{z}(t_0)$ and $\mathbf{z}(t_1)$ corresponds to $x$ and $z$ in Eq. (4) respectively. Since the trace operator $\text{Tr}(\cdot)$ in Eq. (9) preserves permutation invariance, the exchangeability of $\log p(\mathbf{z}(t))$ is maintained along the integral trajectory.

After transformation, we apply a permutation invariant base likelihood to the transformed sets $\mathbf{z}(t)$. For simplicity, we use an i.i.d. base likelihood

$$p_{\mathcal{Z}}(\mathbf{z}(t)) = \prod_{z_i \in \mathbf{z}(t)} p_{\mathcal{Z}}(z_i). \tag{10}$$

The generation process consists of the following steps: 1) Sampling $n$ i.i.d. instances from the base distribution; 2) Inverting the transformations by integrating backwards in time. Although samples from base distribution are independent, the transformations will induce dependencies and transform them to encode global and local structures.

**Training** Like other normalizing flow based models, we train our model by maximizing the log likelihood $\log p_{\mathcal{X}}(\mathbf{x})$ using Eq. (9) and (10). We choose $p_{\mathcal{Z}}(\cdot)$ as $\mathcal{N}(0, I)$ in all our experiments. To reduce memory usage, the adjoint method is used to compute the gradient of a black-box ODE solver [6]. As in FFJORD [19], the trace of Jacobian matrix is estimated using Hutchinson's estimator [21].

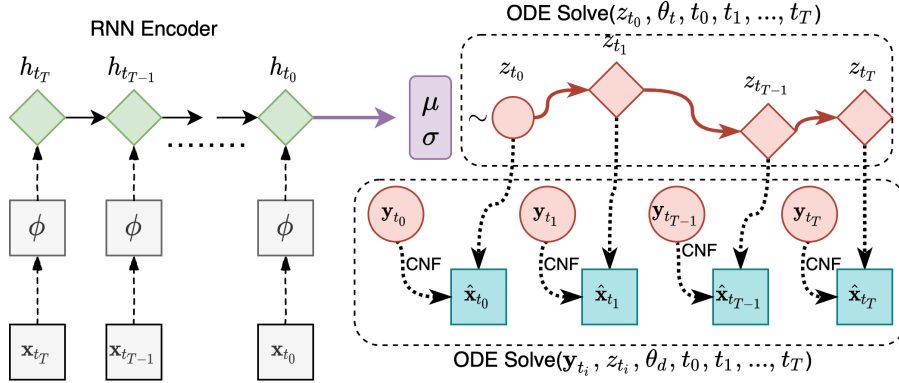

Figure 2: The illustration of encoder and decoder used in temporal set modeling task. The set encoder $\phi(\cdot)$ learns the fix-dimensional permutation invariant representation of a set. The decoder contains two independent ODEs to decode latent states $\mathbf{z}_{t_i}$ and to reconstructed observations $\hat{\mathbf{x}}_{t_i}, i = 0, \dots, T$.

### 3.4 Temporal Set Modeling

In this section, we present a continuous-time VAE model for temporal set modeling. Assume $X = [\mathbf{x}_{t_0}, \mathbf{x}_{t_1}, \dots, \mathbf{x}_{t_N}]$ is a time variant set, where each $\mathbf{x}_{t_i} \in \mathcal{X}^n$ is a set. Let $Z = [z_{t_0}, z_{t_1}, \dots, z_{t_N}]$ be the corresponding latent variables of $X$. We assume that the evolution of latent states can be modeled by an ODE. In other words, given an initial state $z_{t_0}$, other latent states can be inferred following the dynamics $\dot{z}(t)$. Unlike other methods, such as recurrent neural networks (RNNs), where the evaluations can only be performed at prefixed time points, the ODE based model can obtain both the latent states and observations at any time $t$.

Given the latent states $z_{t_i}, i = 0, 1, \dots, T$, we propose to model the conditional distribution, $p(\mathbf{x}_{t_i} \mid z_{t_i})$ using a conditional set CNF. Specifically, the set $\mathbf{x}_{t_i}$ is transformed to a simple base distribution using ExNODE transformations conditioned on the corresponding latent state $z_{t_i}$:

$$\mathbf{x}_{t_i}(s_1) = \mathbf{x}_{t_i}(s_0) + \int_{s_0}^{s_1} g_{\theta_d}(\mathbf{x}_{t_i}(s), z_{t_i}, s)ds, \quad \mathbf{x}_{t_i}(s_0) = \mathbf{x}_{t_i},$$

where $g_{\theta_d}(\cdot)$ defines the transformation dynamics of the CNF in $[s_0, s_1]$. $g_{\theta_d}(\cdot)$ is permutation equivariant w.r.t. $\mathbf{x}_{t_i}(s)$. The log likelihood of $\mathbf{x}_{t_i}$ can be formulated similar to Eq. (9).

**Training** Since computing the posterior distribution $p(z_{t_i} \mid \mathbf{x}_{t_i})$ is intractable, we cannot directly maximize the marginal log likelihood $\log p_\theta(X)$. Therefore, we resort to the variational inference [22, 23] and optimize a lower bound. Following previous work [6, 24] for temporal VAEs, we utilize a recurrent encoder that produces an amortized proposal distribution $\hat{p}_\psi(z_{t_0} \mid X)$ conditioned on the entire time series $X$. The encoder first encodes each set into a permutation invariant representation independently and then uses a recurrent network to accumulate information from each time step. For our models, the encoder processes the time series backwards in time. We assume the prior for $z_{t_0}$ comes from an isotropic Gaussian, $p(z_{t_0}) \sim \mathcal{N}(0, I)$. Latent codes for other time steps are constructed following the dynamics $\dot{z}(t)$. The final encoder-decoder model is illustrated in Fig. 2. We train the encoder and decoder jointly by maximizing the evidence lower bound (ELBO):

$$\text{ELBO}(\theta, \psi) = \mathbb{E}_{z_{t_0} \sim \hat{p}_\psi(z_{t_0}|X)} \left[ \sum_{i=0}^{T} \log p_\theta(\mathbf{x}_{t_i}|z_{t_i}) \right] - \text{KL}(\hat{p}_\psi(z_{t_0}|X)||p(z_{t_0})). \quad (11)$$

**Sampling** After the model is trained, we can sample a set at any time $t$ by first inferring the corresponding latent state $z_t$ and then transforming a set of base samples $\mathbf{y}_t$ conditioned on $z_t$:

$$z_{t_0} \sim p(z_{t_0}), \quad z_t = \text{ODESolve}(z_{t_0}, \theta_t, t) \quad (12)$$

$$\mathbf{y}_t = \{\mathbf{y}_t^j\}_{j=1}^n, \quad \mathbf{y}_t^j \sim \mathcal{N}(0, I), \quad \hat{\mathbf{x}}_t = \text{ODESolve}(\mathbf{y}_t, z_t, \theta_d, t), \quad (13)$$

where $\theta_t$ parametrize the dynamics in the latent-states transmission model and $\theta_d$ parametrize the dynamics of the decoder. Due to the continuous latent space, our model can learn the evolution of sets in time. We can sample sets at unseen time steps by interpolating or extrapolating the latent states.

Table 1: Test Accuracy for point cloud classification with 100 and 1000 points of ModelNet40 dataset. Mean and standard deviation is reported from 5 runs.

| Method | 100pts | 1000pts | # Params |
|---|---|---|---|
| DeepSets [2] | $0.82 \pm 0.02$ | $0.87 \pm 0.01$ | 0.21 M |
| Set Transformer [8] | $0.8454 \pm 0.0144$ | $0.8915 \pm 0.0144$ | 1.15 M |
| ExNODE (deepset block) | $\mathbf{0.8597 \pm 0.0027}$ | $0.8881 \pm 0.0016$ | 0.58 M |
| ExNODE (transformer block) | $0.8569 \pm 0.0015$ | $\mathbf{0.8932 \pm 0.004}$ | 0.52 M |

## 4 Experiment

The experiments are divided into three parts. First, we evaluate ExNODE on point cloud classification (Sec. 4.2). Second, we conduct experiments to validate the efficacy of ExNODE for point cloud generation and likelihood estimation (Sec. 4.3). Finally, we explore the temporal set modeling task (Sec. 4.4), where interpolated and extrapolated samples are generated to demonstrate the benefits of the continuous-time model. Our implementation of neural ODE utilizes the official implementation of the NODE [6]. We post our code at `https://github.com/lupalab/ExNODE`.

### 4.1 Architecture

We consider two different exchangeable base architectures in constructing an ExNODE model: one based on DeepSets [2] and the other on Set Transformers [8].

DeepSets provides both necessary and sufficient conditions for implementing permutation equivariant functions. In practice, independent element-wise and pooling operations are used to preserve equivariance and capture dependencies, i.e. $f(\mathbf{x}) = \sigma(\lambda I \mathbf{x} + \gamma \, \text{pool}(\mathbf{x}))$.

Recently, the attention-based *Transformer* has remarkably boosted performance in natural language processing since the transformer can encode pair-wise interactions between elements in sequential data [25]. Set Transformer extends the transformer architecture to sets by defining self-attention based operations over set elements. Using self-attention mechanism comes with several advantages: 1) pair-wise interactions are explicitly modeled; 2) stacking multiple blocks can capture higher-order interactions.

The use of Neural ODE framework does add overhead for ExNODE. Depending on the solver used, the running time can be quite different. `RK4` solver is considerably faster than adaptive solvers like `dopri5`, but sometimes it leads to numerical issues. We use `dopri5` for flow models and `RK4` for classification models. The generative flow models could take roughly 4 days on one TITAN XP GPU, while the classification converges within couple hours (Shown in Fig. 3).

### 4.2 PointCloud Classification

In this section, we evaluate ExNODE on point cloud classification using ModelNet40 [26], which is composed of surface points from 40 different categories of 3D CAD models. We train our model with randomly sampled 100 and 1000 points, respectively. Since ExNODE can easily utilize different exchangeable blocks to learn set representations, we train our model using both DeepSets and Set Transformer blocks. The test classification accuracy of different models are shown in Table 1. We report the mean and standard deviation from 5 runs with different random seeds. For both small sets (100pts) and large sets (1000pts), ExNODE consistently outperforms the baselines. ExNODE additionally achieves better performance in terms of parameter efficiency, requiring approximately half the number of parameters compared to Set Transformer and still achieving superior performance.

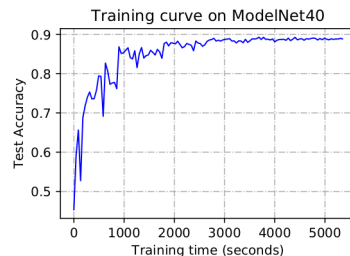

Figure 3: The classification accuracy along training on ModelNet40 using 1000 points. `RK4` solver is used and the training converges quickly.

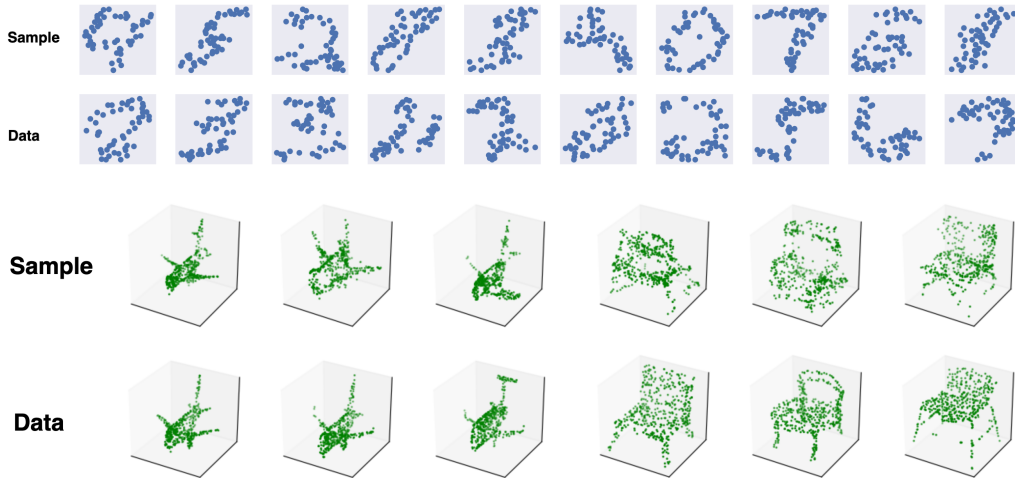

Figure 4: Generated samples and real data for SpatialMNIST (top) and ModelNet40 (bottom). SpatialMNIST consists of 50 points per set, and ModelNet40 contains 512 points.

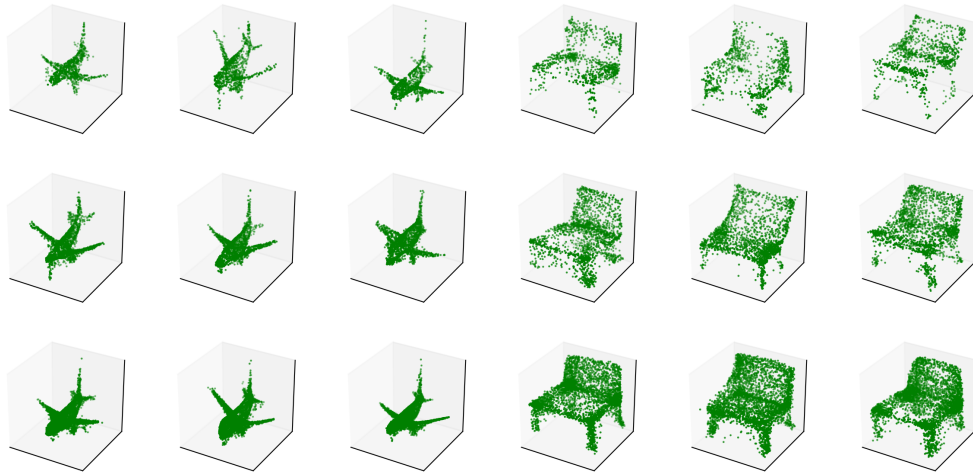

Figure 5: From top to bottom, we show sampled sets with 1024, 2048 and 4094 points, respectively. The model is trained with 512 point.

## 4.3 Set Generation and Density Estimation

Next, we conduct experiments for set generative task using *SpatialMNIST* [3] and *ModelNet40* [3]. *SpatialMNIST* consists of 50 2d points sampled uniformly from active pixels of MNIST. *ModelNet40* are constructed by sampling 512 points uniformly from one category. Architectural details are provided in Appendix D. The per-point log likelihood (PPLL) from the trained ExNODE and other baselines can be found in Table 2. ExNODE outperforms other models in all three datasets. Figure 4 shows the generated samples from our model. Although our model is trained with only 512 points, it is capable of generating more points by sampling more points from the base distribution. Figure 5 shows some examples for *airplanes* and *chairs*.

## 4.4 Temporal Set Modeling

For the set temporal generation task, we also use *SpatialMNIST*. To generate a temporal set series, we clockwise rotate the digits of MNIST dataset in a constant speed from $t = 0$ to $t = 1$ and

Table 2: Per Point Log Likelihood (PPLL) on test set. Higher is better.

| Dataset | BRUNO [4] | NS [3] | FlowScan [5] | ExNODE |
|---|---|---|---|---|
| Chairs | 0.75 | 2.02 | 2.58 | **3.59** |
| Airplanes | 2.71 | 4.09 | 4.81 | **5.13** |
| SpatialMNIST | -5.68 | -5.37 | -5.26 | **-5.21** |

sample 50 points from the active pixels at random. We train our model at five *fixed* time points, $t = [0, 0.25, 0.5, 0.75, 1]$. For details about the architectures, please refer to appendix D. Given the initial latent state, $z_0$, ExNODE can generate latent state at any time and then generate corresponding samples conditioned on the latent state. As shown in Fig. 6, we can both interpolate and extrapolate to unseen time steps. The samples generated at interpolated time maintain the smoothness over time. See Appendix B for conditional samples where $z_0$ is encoded by a given series as a reconstruction. Interpolation on latent code $z_0$ suggests our model learns a meaningful latent space.

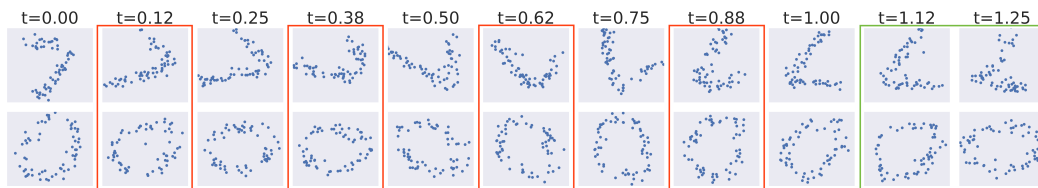

Figure 6: Samples form the temporal VAE. Red boxes indicate interpolated time steps, and green boxes indicate extrapolated time steps.

## 5 Conclusion

In this work, we extend neural ODEs to model exchangeable sets. We prove that the solution of an ODE $\dot{x}(t) = f(x(t), t)$ is permutation equivariant w.r.t. initial value through time as long as its first order derivative $f(x(t), t)$ is permutation equivariant w.r.t. $x(t)$. Therefore, we parameterize $f(x(t), t)$ as a permutation equivariant neural network and use black-box ODE solver to find the solution. Since the ODE block is naturally invertible, we can apply our ExNODE in a flow based generative model as an equivariant flow transformation. According to the CNF formulation, we can compute the likelihood by solving an ODE. We also propose to model the time variant sets using a continuous-time VAE model. We observe smooth transition along time at both interpolated and extrapolated time steps. In future works, we will evaluate on other applications, such as traffic tracking.

## Broader Impact

Making assessment over sets instead of instances gives us opportunity to leverage the dependencies over set elements. However, like any other models, it might unintentionally exploit the bias within the dataset. With this known issue, we encourage practitioners to carefully design the training set or utilize other debiasing techniques. In this work, we evaluate on point clouds of shape objects, which should not pose detrimental societal impact even if the learned dependencies does not reflect the actual ones.

Set generative models have the ability to generate fake data, which may incur ethical or legal issues when used improperly. There is urgent need to establish regulations and techniques to avoid misuse of the generated data.

## Acknowledgments and Disclosure of Funding

This work was supported in part by NIH 1R01AA02687901A1.

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
