[Supplementary Material]

## A   Proofs

**Theorem 1**   (Permutation Equivariant ODE) Given an ODE $\dot{z}(t) = f(z(t), t), z(t) \in \mathcal{X}^n$ defined in an interval $[t_1, t_2]$. If function $f(z(t), t)$ is permutation equivariant w.r.t. $z(t)$, then the solution of the ODE, i.e., $z^\star(t), t \in [t_1, t_2]$ is permutation equivariant w.r.t. the initial value $z(t_1)$. We call the ODE with permutation equivariant properties as ExODE.

*Proof.* For any permutation $\pi(\cdot)$, we have

$$\pi(z^\star(t)) \stackrel{(1)}{=} \pi(z(t_1)) + \pi(\int_{t_1}^{t} f(z(\tau), \tau)d\tau)$$

$$\stackrel{(2)}{=} \pi(z(t_1)) + \int_{t_1}^{t} \pi(f(z(\tau), \tau))d\tau$$

$$\stackrel{(3)}{=} \pi(z(t_1)) + \int_{t_1}^{t} f(\pi(z(\tau)), \tau)d\tau$$

$$\stackrel{(4)}{=} g(\pi(z(t_1)), f, t)$$

$\square$

## B   Temporal Set Modeling

Figure 6: Additional samples from our temporal VAE.

## C   Training details

### C.1   Point cloud classification

The details of network architecture we used are shown in Table 3. All the models are trained using Adam optimizer with $\beta_1 = 0.9$ and $\beta_2 = 0.999$. The learning rate and batch size are set to 1e-3 and 64 in all experiments. We use the fourth order Runge-Kutta solver to solve the ExNODE in our

Figure 7: Conditional samples using the encoded $z_0$ of the first row.

Figure 8: Interpolate $z_0$ from two different temporal sets (the first and the last row).

model, and the numeric tolerance is set to 1e-5 in all experiments. We train our model on a single NVIDIA Tesla V100 GPU. For generalization, we randomly rotate and scale each set during training with $n = 1000$ points.

## C.2 Set generation

The details of network architecture are provided in Table 3. The batch is set to 128 in all experiments. We train our model using Adam optimizer with an initial learning rate of 1e-3 which we decay by a factor of 0.5 every 100 epochs. We use `dopri5` solver to solve the ODE with numeric tolerance of 1e-5.

### C.3 Set temporal model

The dimension of the latent state variable $z_0$ is set to 128. We randomly sample 64 points uniformly from active pixels of MNIST dataset as a set. We train our model using Adam optimizer with learn rate 1e-3, $\beta_1 = 0.9$ and $\beta_2 = 0.999$, respectively. The batch size is set to 128 in all experiments. We use `dopri5` solver to solve the ODE used in our model, and the relative and absolute numeric tolerance are set to 1e-3 and 1e-4, respectively. All the models are trained on a single NVIDIA Tesla V100 GPU.

**Decoder**    The decoder models the reconstruction likelihood $p(\mathbf{x}_{t_i}|z_{t_i})$. We share the same decoder at different time. The `concatsquash`-like linear layers are used in our CNF decoder:

$$CCS(\mathbf{x}, z, t) = (W_x\mathbf{x} + b_x) * \text{gate} + \text{bias},$$

where $\text{gate} = \sigma(W_{tt}t + W_{tz}z + b_t)$ and $\text{bias} = W_{bt}t + W_{bz}z + b_bt$. In our experiment, we stack four `concatsquash` linear layers to model the dynamics $g_{\theta_d}$. We also use `Tanh` activation to connect the consecutive `concatsquash` linear layers. For more details of network architecture used in our model, see Table 3.

## D   Architecture

See next page.

Table 3: Detailed network architecture used in our experiments for different tasks.

| Model | Dataset | Architecture | |
|---|---|---|---|
| PointCloud Classification (deepset block) | ModelNet40 | Input | $64\times3\times100$ or 1000 |
| | | FE | Conv1d 64x1(stride 1) BN(64) Tanh |
| | | | Conv1d 256x1(stride 1) BN(256) Tanh |
| | | ExNODE | FC (512) Tanh FC(512) FC(256) |
| | | Pooling | Max(1) Flatten |
| | | Prediction | FC(128) BN(128) Tanh FC(40) |
| PointCloud Classification (transformer block) | ModelNet40 | Input | $64\times3\times100$ or 1000 |
| | | FE | Conv1d 64x1(stride 1) BN(64) Tanh |
| | | | Conv1d 256x1(stride 1) BN(256) Tanh |
| | | ExNODE | K: FC(256) Tanh FC(256) |
| | | | Q: FC(256) Tanh FC(256) |
| | | | V: FC(256) Tanh FC(256) |
| | | | FC(256) |
| | | Pooling | Max(1) Flatten |
| | | Prediction | FC(128) BN(128) Tanh FC(40) |
| Set Generation | SpatialMNIST | Input | $128\times50\times2$ |
| | | ExNODE $\times12$ | K: FC(128) Tanh FC(128) Tanh FC(128) |
| | | | Q: FC(128) Tanh FC(128) Tanh FC(128) |
| | | | V: FC(128) Tanh FC(128) Tanh FC(128) |
| | | | FC(2) |
| Set Generation | ModelNet40 | Input | $128\times512\times2$ |
| | | ExNODE $\times12$ | K: FC(128) Tanh FC(128) Tanh FC(128) |
| | | | Q: FC(128) Tanh FC(128) Tanh FC(128) |
| | | | V: FC(128) Tanh FC(128) Tanh FC(128) |
| | | | FC(3) |
| Temporal Set Model | SpatialMNIST | Input | $128\times64\times2$ |
| | | Encoder($\phi$) | Conv1d 128x1(stride 1) BN(128) ReLU |
| | | | Conv1d 128x1(stride 1) BN(128) ReLU |
| | | | Conv1d 256x1(stride 1) BN(256) ReLU |
| | | | Conv1d 512x1(stride 1) BN(512) Max |
| | | RNN | GRU(513, 512) (Concat $\Delta t$ as Input) |
| | | RNN_to_$z_0$ | mean: FC(256) BN(256) RELU |
| | | | FC(128) BN(128) RELU FC(128) |
| | | | std: FC(256) BN(256) ReLU |
| | | | FC(128) BN(128) ReLU FC(128) Exp |
| | | Latent: $z_0$ | 128 |
| | | ODE($z_t$) | FC(256) Tanh FC(256) Tanh FC(128) |
| | | ODE($\hat{\mathbf{x}}_t$) | Concatsquash Linear $\times$ 4: |
| | | | 1) FC(2, 512) gate: FC(129, 512, bias=F) |
| | | | bias: FC(129, 512) (Concat $t$ and $z$) |
| | | | 2) FC(512, 512) gate: FC(129, 512, bias=F) |
| | | | bias: FC(129, 512) (Concat $t$ and $z$) |
| | | | 3) FC(512, 512) gate: FC(129, 512, bias=F) |
| | | | bias: FC(129, 512) (Concat $t$ and $z$) |
| | | | 4) FC(512, 2) gate: FC(129, 2, bias=F) |
| | | | bias: FC(129, 2) (Concat $t$ and $z$) |