[Reviews · NeurIPS 2020]

Review 1

Summary and Contributions: UPDATE: I've read the rebuttal and it addressed all my concerns. I still think it's a good paper and recommend acceptance. ### This paper has three contributions: - It proposes using permutation-equivariant drift functions in the neural ODE framework for modelling sets, and evaluates them in a discriminative setting on point-cloud classification achieving SOTA performance. - It uses neural ODEs with a permutation-equivariant drift function to parametrize a continuous normaliing flow, which results in a permutation-equivairant generative model with a permutation-invariant likelihood. This model achieves SOTA likelihoods on point-clouds. - It introduces a VAE for modelling sets that change in time, where the decoder is realized as the above CNF, while the latent-state transition obey the latent neural ODE framework. This is not compared to any baselines.

Strengths: I find the paper quite strong, and specifically: - the introduced framework is theoretically sound; the authors describe constraints that must be met when modelling sets (taking dependencies between points into account, permutation-equivariance and -invariance), and shows that the introduced method does meet them. - The empirical evaluation is sufficient for the first two tasks, that is classification and generative set modelling, which are perhaps more important than the last task of temporal set evolution. - There has been growing interest in the community in set modelling, with several other NeurIPS submissions and several papers at last ICML. This paper fits very well in this line of work, and proposes a novel (up to combination of existing methods, that is perm-equiv transformations, neural ODEs and continuous normalizing flows), and provides a new important tool that can be useful for applications considering modelling sets.

Weaknesses: - Even though the work is theoretically sound, and the authors do mention that drift functions used in the neural ODE framework have to be Lipschitz continuous, there is no mention if and how Lipschitz continuity can be achieved in general, and if it is achieved for the specific parametrizations (deep set, set transformer) used in this work. More specifically, the vanilla self-attention of (set-)transformer is provably NOT Lipschitz continuous as shown in Kim et. al, "The Lipschitz Constant of Self-Attention". Now, I do realize that the paper I am referencing is an arxiv submission that appeared after the NeurIPS application deadline, and I do not hold this against the authors. However, the authors do not even mention that there was no/authors were not aware of any results on Lipschitz continuity of the used modules. Moreover, deep set need not be Lipschitz continuous if no measures to ensure it are taken. To sum up, I would like the authors to include a paragraph of discussion on the topic--this does not diminish the contributions of this paper! - The paper does not mention how computationally expensive the proposed methods are. We know from the previous neural ODE papers that they are extremely expensive (FFJORD used about 8 gpus for a week to train a model for MNIST classification (sic!)); an unfamiliar reader might be mislead and think that this is an efficient method. I would encourage the authors to both explicitly comment on this issue, as well as provide training curves showing performance vs wall-clock time. - The temporal set modelling method and experiment seem somewhat contrived. -- First, I do recognize the advantages of interpolation between timesteps granted by ODE evolution of the latent state, but having only z_0 as a stochastic variable means that this will not work in stochastic environment (stochastic state transitions), and it makes inference unnecessarily difficult--the whole sequence has to be compressed into a single vector! -- Second, the provided experiment has no baselines (you could compare it with e.g. Gregor et. al., "Temporal Difference Variational Auto-Encoder", or at least a VRNN of Chung et. al., "A Recurrent Latent Variable Model for Sequential Data" conditioned on time-step or time increment, but I am sure that other methods are available). -- Third, the presented qualitative results are not that convincing: is it just me, or do the "extrapolation results" in Fig 5 look quite a bit distorted compared to earlier timeframes? -- Finally, L192 mentions that the posterior distribution is intractable. Strictly, this is not true generally, but only in the considered setting due to a modelling choice. If z and x had the same dimensionality, then all zs and xs could be decoded from z_0 through ODE evolution starting from z_0. I do recognize that in the presented setting the posterior is, in fact, intractable, but it would be nice to have a discussion similar to mine in the paper.

Correctness: Everything seems to be correct, minus the Lipschitz continuity aspect discussed in the above section.

Clarity: The paper is very clear and well written. - L212 has a type in "benefits" - L248: is it 50 point sampled initially and transformed, or do you transform all points and sample 50 points at every timestep independently? This is unclear.

Relation to Prior Work: Yes, related work is very clearly discussed. I would only add in L74-82 that although deep sets are universal approximators, there are some constraints wrt the dimensionality of latent representation used as shown by Wagstaff et. al., "On the limitations of representing functions on sets"

Reproducibility: Yes

Additional Feedback:


Review 2

Summary and Contributions: The paper proposes a model architecture for sets that obey permutation equivariance via parameterising the drift function of a neural ODE with a permutation equivariant neural net, such as DeepSets or the Transformer. This is used for both discriminative tasks (point cloud classification) and generative tasks (point cloud generation/density estimation).

Strengths: 1. The paper extends the scope of Neural ODEs to modelling sets, and as far as I know is one of the first work to use the Transformer with a neural ODE. 2. The method is simple, and it works. 3. The paper is written clearly and presented in a way that is easy to follow 4. The evaluation (while being restricted to point clouds) covers both discriminative and generative tasks.

Weaknesses: The design choices of the temporal set model is not very well explained. 1. It’s not clear to me why you would want to model p(x_t|z_t) again with a neural ODE. Intuitively it seems sufficient to model p(z_t) using a neural ODE, then p(x_t|z_t) with a (conditional) permutation equivariant decoder (e.g. DeepSets or Set Transformer). Using a neural ODE for p(x_t|z_t) seems to pose an unnecessary burden on the time/memory required by the implementation, and to justify this an ablation study that compares different choices of p(x_t|z_t) would be helpful. 2. Using an RNN encoder will only make sense if the data comes at fixed time points [t_0,...,t_N]. Hence this approach would be unable to exploit the benefit of the ODE that can model dynamics at any t. Have you tried using a neural ODE for the encoder as well? 3. How does the VAE model for temporal set modelling compare to just using a single CNF to model x_t for all times t? i.e. you would optimise \sum_{i=1}^T log p(x_t_i) where log p(x_t_i) = log p(x_t_{i-1}) - \int_{t_i}^{t_{i-1}} Tr(df/dx(t)) dt. This would avoid having to use a decoder, and seems to be a much more natural approach to modelling temporal dynamics with a neural ODE. Also the results for the temporal set model are not very convincing: 4. The claim that the ExNODE can extrapolate to unseen time steps for the temporal model seems unjustified. In Figure 5 it looks like it’s struggling to capture the rotation for t > 1 5. The samples don’t look rotation invariant (i.e. the relative positions of the points appear to change with time) When comparing to other baselines, there are no mentions of training times. I believe one weakness of neural ODEs is that they are slow, and so it would be helpful to show how much slower they are compared to DeepSets and Set Transformer, and what would be the implications for scaling up to bigger datasets e.g. Do you expect the temporal model to be feasible to train for rotating ModelNet40 point clouds? Also how does the method compare to PointNet(++), which is a strong baseline for point cloud classification?

Correctness: The paper has no issues on correctness, as far as I can tell.

Clarity: The paper is clear in its presentation of the methods and experimental results.

Relation to Prior Work: It is clearly stated how the work differs from prior work.

Reproducibility: Yes

Additional Feedback: Edit after rebuttal: the rebuttal addressed about half of the points mentioned in the review, but the remaining half remains unanswered. It would be good to discuss the remaining points in the revised version of the paper. But given the tight time/space constraints for the rebuttal, I agree that the authors have made an effort to address the concerns in the review, so will raise the score by one.


Review 3

Summary and Contributions: The paper marries two recent concepts: Neural ODEs on the one hand, and DeepSet architectures on the other hand. By using equivariant neural set architectures in the neural ODEs, they arrive at a new flavor of permutation-invariant DeepSet architecture. This architecture is subsequently used for discriminativ as well as generative models of sets.

Strengths: The paper presents a clean, straight-forward, yet non-trivial combination of two interesting neural architectures of the last years. The realm of set architectures is still in its infancy, and the suggested approach bears promise in that it can benefit from advances in either of the two areas it marries. The suggested avenue is particularly promising because it seamlessly bridges the gap between discriminative and generative tasks. Specifically the contribution to the latter is interesting. The results here are very promising.

Weaknesses: The classification results are over-stated. One should not forget that the suggested model is after all a variant of (or using!) the baselines. SotA results are hence more of a sanity check rather than an achievement. This shows in the numbers of Table 1: The Set Transformer performance is within the margin of error of ExNODE. This is not a blow against ExNODE, merely against the presentation and claims. Each method has its relative merits, and those should be emphasized more. The table shows a reduced number of parameters (does the performance increase with a number of parameters equal to Set Transformer?). At the same time, I would expect the computation time of ExNODE to be more costly. Given the results in the generative modeling sections, there is no need to overstate the results. The notion of capturing intradependencies is emphasized, but never quite proven. The experimental section does not back this up. It's a natural hypothesis---but it remains a hypothesis. The paper should reflect that better, either by toning down or by adding experiments making the claim palpable.

Correctness: Both are correct.

Clarity: The paper is well-structured and easy to follow. Minor typos etc.: "time-variant", "time-invariant", "permutation-equivariant" and the like should be hyphenated. 192: Traning 257: The non-bold notation seems out of sync with the method section.

Relation to Prior Work: I do not have a very good global overview about the recent state of the literature, but I'm not missing key references. The related work is discussed rather neutrally. It is hard to actually relate the related work to the current approach and understand the differences and relative merits.

Reproducibility: Yes

Additional Feedback: I believe the overall idea is good and worth publishing, even as is. The method has legs and room for improvement later. Still, I believe the paper could be even better if it were a bit more open about relative advantages and disadvantages towards competing methods---be confident and present interesting results, not good results. A minor suggestion: A paper is obviously a bad medium for this, but it would be interesting to see the interpolated and extrapolated temporal generations in an animation. A sequence of images makes it impossible to detect glitches, non-smooth behavior, or discrepancies towards the ground truth. This would have been great supplementary material. --------- POST-REBUTTAL COMMENTS In my review, I raised the point "The notion of capturing intradependencies is emphasized, but never quite proven." to which you answered "Intuitively, the permutation equivariant network that parametrizes the drift function should be able to learn the intradependencies." To me, that proves my point rather than answering it---I am not asking for intuition, I am asking for evidence. I would recommend removing all related paragraphs from the final version. While the rebuttal leaves a sour taste in this regard, it's not enough to change my rating.


Review 4

Summary and Contributions: The paper proposes a novel architecture for modeling sets, also including sets that vary in time. The models are suitable for classification and generation tasks, where they achieve very good performance. The models are based on neural ODEs with permutation-equivariant neural networks, which is an interesting combination.

Strengths: The paper makes sound claims and I think it's a valuable addition to the group of set models. With NeuralODEs growing more popular, the proposed model might be relevant to the community if not in its present form, then as a source of inspiration for other models.

Weaknesses: There are some statements in the paper that I'm uncertain of. Firstly, in several places, the paper says that simple approaches process instances independently, while exNODE explicitly captures intradependencies among set elements. If under simple approaches, the paper means DeepSets and Neural Statistician, then I would disagree, esp. considering that the form of the neural network used in exNODE is the same as in DeepSets. Regarding the Neural Statistician, it assumes conditional independence of the set elements given some latent variable, which is fully justified by de Finetti's theorem. Since there are different ways to model interdependencies, I wouldn't belittle them in comparison to exNODE. Secondly, while I'm fond of neural ODEs, I would still like the paper to give a better motivation for their use. For example, why not use other flow-based models? Thirdly, I'm a bit unhappy with how the word 'exchangeable' is used. It has a particular meaning in stochastic processes literature, and the way it is re-defined here might be confusing. So I would suggest to use 'permutation-invariant' instead of 'exchangeable'. Lastly, the paper has no discussion on the limitations of the proposed method. I have a few guesses: 1) the model can only be trained on sets with a fixed size 2) it might not work well on sets of high dimensional points, while BRUNO and Neural Statistician (cited in Table 2) are good at it. Also, I think it would be interesting to compare exNODE with Neural Processes, which are currently very popular generative models of exchangeable sequences. I would gladly increase my current score if the above points are properly addresses.

Correctness: yes

Clarity: there are some unclear parts, but mostly it's okay

Relation to Prior Work: yes

Reproducibility: Yes

Additional Feedback: Some random comments: -- it would be useful to have eq. for f(x) in line 219 earlier in the paper, e.g. around line 142 -- I wonder if images in Figure 4 can be made consistent. For instance, 2048 points in the second row are preserved in the image with 4094 points. -- line 92: 'exchangeable sets' -> just 'sets' or 'exchangeable sequences'. Though, again, I'm not a big fan of how the concept of exchangeability is used is the paper. -- line 40: I think that all flow models are invertible, so I would rephrase the sentence which says that it's an additional requirement. ---------------- POST-REBUTTAL UPDATE ------------------ The rebuttal did not answer all of my questions, probably due to the lack of space and time. Thus, I'll be looking forward to the final version of the paper that will include the discussion of the limitations. I have increased my score to 7.

[Author Response · NeurIPS 2020]

We would like to thank the reviewers for their time and helpful notes.

**General Comments:**

The use of Neural ODE framework does add overhead for ExNODE. Depending on the solver used, the running time can be quite different. RK4 solver is considerably faster than adaptive solvers like dopri5, but sometimes it leads to numerical issues. We use dopri5 for flow models and RK4 for classification models. The generative flow models could take roughly 4 days on one TITAN XP GPU, while the classification converges within couple hours (Shown in Fig. 1). We will add some comments on this issue in the camera-ready version.

Figure 1: The testing accuracy over training time on Model-Net40 using 1000 points.

The temporal set modeling is a novel exploratory task, which could be an interesting future direction. It also has impactful applications such as modeling the traffic flow. As pointed out by Reviewer #2, the RNN encoder cannot deal with irregular time steps; thus other temporal architecture may be of use. The extrapolation is indeed harder than interpolation. We suspect that is because the VAE model is not trained to generalize beyond the seen time steps.

**Reviewer 1:**

Please refer to general comments for discussions about run time and temporal experiments.

Thank you for pointing out this very interesting paper. The Lipschitz continuous constraint is an important factor and we will add some discussion in the camera-ready version.

It is a good idea to use latent code with the same dimension. We experimented with exchangeable latent codes, where $z$ is another set with the same cardinality encoded from $x$ using ExNODE. However, we found it hard to learn and the generated samples do not look good. We will inspect into this issue in the future work.

L74-82: We will add the description about constraints for deepset.

L248: We rotate the image and then sample 50 points at each time step independently.

**Reviewer 2:**

Please see general comments for discussions about run time and temporal experiments.

As you said, the decoder for $p(x_t \mid z_t)$ can use other architectures, like deepsets or set transformer, but it will require the use of distance-based objectives, like the earth mover distance or Chamfer distance. Here, we employ the ExNODE based generative flow model to simplify training so that we can directly maximizing the conditional likelihood.

The method you describe seems like a particle flow model, which is interesting and was considered in the early stage. We found that learning temporal correspondence over sets is surprisingly difficult. We will inspect into this issue in the future work.

For ModelNet40 with 1000 points, ExNODE gets 89.32, while PointNet++ gets 90.7, which is close, however ExNODE uses much fewer parameters.

**Reviewer 3:**

Please see general comments for discussions about run time.

Intuitively, the permutation equivariant network that parametrizes the drift function should be able to learn the intradependencies. Showing an animation is a good idea and would provide further insights. We will add one in the supplementary material for camera-ready version.

**Reviewer 4:**

One advantage of using continuous normalizing flow for set modeling is its invertibility. We do not need to design special structures, like coupling transformation, to guarantee invertible. We can basically use any architecture as long as they are permutation equivariant.

We will add some discussions about limitations of ExNODE in the camera-ready version. As shown in general comments, the computation would be one of the limitations. It would be even more expensive for high dimensional sets, like sets of images.

[Meta-Review · NeurIPS 2020]

Good paper that develops continuous-time normalizing flows for sets, by building in the associated permutation symmetry. Good contribution, that extends the field of normalizing flows into the direction of explicit incorporation of known symmetries. Well done. One concern the reviewers raised was that the Transformer-parameterized vector field may not be Lipschitz. I would encourage the authors to acknowledge this point in the camera-ready, and discuss whether we would expect this to be an issue or not.